# Murine Oncostatin M Has Opposing Effects on the Proliferation of OP9 Bone Marrow Stromal Cells and NIH/3T3 Fibroblasts Signaling through the OSMR

**DOI:** 10.3390/ijms222111649

**Published:** 2021-10-28

**Authors:** Lena Jakob, Tony Andreas Müller, Michael Rassner, Helen Kleinfelder, Pia Veratti, Jan Mitschke, Cornelius Miething, Robert A. J. Oostendorp, Dietmar Pfeifer, Miguel Waterhouse, Justus Duyster

**Affiliations:** 1Department of Hematology and Oncology, Freiburg University Medical Center, Albert-Ludwigs-University of Freiburg, Hugstetter Str. 55, 79106 Freiburg, Germany; lena.jakob@uniklinik-freiburg.de (L.J.); tony.mueller@uk-koeln.de (T.A.M.); Michael.rassner@uniklinik-freiburg.de (M.R.); helen.kleinfelder@uniklinik-freiburg.de (H.K.); pia.veratti@uniklinik-freiburg.de (P.V.); Jan.mitschke@uniklinik-freiburg.de (J.M.); Cornelius.miething@uniklinik-freiburg.de (C.M.); dietmar.pfeifer@uniklinik-freiburg.de (D.P.); miguel.waterhouse@uniklinik-freiburg.de (M.W.); 2Center for Integrated Oncology (CIO), Department I of Internal Medicine, Aachen-Bonn-Cologne-Duesseldorf, Excellence Cluster for Cellular Stress Response and Aging-Associated Diseases (CECAD), Center for Molecular Medicine Cologne (CMMC), University of Cologne (UoC), 50937 Cologne, Germany; 3German Cancer Consortium (DKTK) and German Cancer Research Center (DKFZ), 69120 Heidelberg, Germany; 4Department of Internal Medicine III, Technical University of Munich, Klinikum Rechts der Isar, 81675 Munich, Germany; robert.oostendorp@tum.de

**Keywords:** mOSM, proliferation, JAK-STAT, OSMR, LIFR

## Abstract

The IL-6 family cytokine Oncostatin M (OSM) is involved in cell development, growth, hematopoiesis, inflammation, and cancer. Intriguingly, OSM has proliferative and antiproliferative effects depending on the target cell. The molecular mechanisms underlying these opposing effects are not fully understood. Previously, we found OSM upregulation in different myeloproliferative syndromes. However, OSM receptor (OSMR) expression was detected on stromal cells but not the malignant cells themselves. In the present study, we, therefore, investigated the effect of murine OSM (mOSM) on proliferation in stromal and fibroblast cell lines. We found that mOSM impairs the proliferation of bone marrow (BM) stromal cells, whereas fibroblasts responded to mOSM with increased proliferation. When we set out to reveal the mechanisms underlying these opposing effects, we detected increased expression of the OSM receptors OSMR and LIFR in stromal cells. Interestingly, *Osmr* knockdown and *Lifr* overexpression attenuated the OSM-mediated effect on proliferation in both cell lines indicating that mOSM affected the proliferation signaling mainly through the OSMR. Furthermore, mOSM induced activation of the JAK-STAT, PI3K-AKT, and MAPK-ERK pathways in OP9 and NIH/3T3 cells with differences in total protein levels between the two cell lines. Our findings offer new insights into the regulation of proliferation by mOSM.

## 1. Introduction

The IL-6 family cytokine Oncostatin M (OSM) was first described as a factor released by U937 lymphoma cells with the ability to inhibit the growth of the human melanoma cells A375 in vitro [1]. Numerous studies have since revealed its role in different physiological and pathological processes such as cell growth [1], development [2,3], inflammation [4], hematopoiesis [2,5,6], and cancers [7,8,9,10,11,12,13].

Physiologically, OSM is released by activated leukocytes including macrophages [1,14], monocytes [15], T-lymphocytes [16,17], dendritic cells [18] and neutrophils [19,20,21,22,23]. Similar to other IL-6 family cytokines, OSM affects target cells by binding to membrane receptors consisting of two subunits: the transmembrane protein gp130 and a ligand-specific subunit [24,25]. Human OSM (huOSM) activates a type I OSM receptor, composed of gp130 and leukemia inhibitory factor receptor (LIFR), or a type II OSM receptor composed of gp130 and OSMR [26,27], whereas murine OSM (mOSM) mainly interacts with the type II OSM receptor [28,29]. However, there is also evidence for mOSM signaling involving both type I and II OSM receptors in bone formation and resorption [30,31]. LIFR is expressed by various cell types including hematopoietic cells, while the expression of OSMR is restricted to mesenchymal cells like fibroblasts, endothelial cells, osteoblasts, and epithelial cells, and a number of cancer cells [32].

Generally, the cellular responses and downstream pathway activation after OSM binding to its receptors are highly dependent on the target cell type. Activation of type I and II receptors by OSM induces downstream pathways such as JAK-STAT, MAPK-ERK, and PI3K-AKT [33,34,35]. More recent studies revealed activation of c-Jun N-terminal kinase (JNK), p38 mitogen-activated protein kinase [36] and protein kinase C-delta (PKCd) pathways by OSM [37,38,39].

Although OSM was initially described as an antiproliferative cytokine inhibiting the growth of melanoma, breast cancer, chondrosarcoma, lung adenocarcinoma, and glioblastoma cells [1,40,41,42,43], it has also been shown to promote proliferation in various cell lines [44,45,46,47,48,49]. The mechanisms of these differential effects on certain cell types are not fully understood yet.

In previous studies, OSM was found to be upregulated in different myeloproliferative syndromes [50,51]. Its receptor OSMR however is not expressed on the malignant cells themselves but can be found on bone marrow stromal cells [6,52]. This indicates that stromal cells as part of the tumor microenvironment might be involved in tumor progression. It is known that aberrant proliferation and interaction of tumor cells with their microenvironment are important part aspects of tumorigenesis [53,54].

In the present study, we, therefore, investigated the molecular mechanisms leading to the opposing effects of OSM on proliferation in murine bone marrow stromal cells and fibroblasts.

## 2. Results

### 2.1. OSM Has Differential Effects on the Proliferation of BM Stromal Cells and Fibroblasts

Based on its upregulation in various cancer entities [7,8,9,10,11,12,13] and its controversial role in tumorigenesis, we investigated the differential effects of OSM on the proliferation of stromal cells and fibroblasts. In an initial experiment, we treated the murine BM stromal cell lines OP9, and M2-10B4, the murine fetal liver stroma cell line EL08-1D2 as well as primary BM stromal cells with OSM and analyzed the cell cycle progression and proliferation using an EdU incorporation assay. The proliferation of OP9 (Figure 1B), EL08-1D2 (Appendix A) and M2-10B4 (Appendix A) was impaired by mOSM, indicated by a reduced percentage of cells undergoing the S phase, while the percentage of cells remaining in the G0/G1 phase was increased. The inhibitory effect of mOSM was most pronounced for OP9 cells demonstrating OSM-induced reduction of S phase percentage by 52%, compared to S phase percentage reduction of 19% in EL08-1D2 and 30% in M2-10B4 cells. Consistent with the effect of OSM on the BM stromal cell lines, OSM treatment reduced the S phase percentage of murine primary stromal cells by 68% (Figure 1A) and human primary stromal cells by 28% (Appendix A).

It has previously been demonstrated that mOSM promotes the proliferation of NIH/3T3 fibroblasts [55]. We confirmed that OSM-exposed NIH/3T3 cells presented with a 2.8-fold increase of S phase percentage compared to untreated cells. In contrast to OP9 cells, NIH/3T3 cells showed a reduction of cells remaining in the G0/G1 phase after mOSM treatment (Figure 1C). OP9 and NIH/3T3 cells are known to proliferate fast. To detect and compare the effects of mOSM on proliferation cell lines were starved for all experiments.

### 2.2. OP9 and NIH/3T3 Cells Show Differential Levels of OSMR and LIFR

To understand the mechanism leading to the differential effects on proliferation, we investigated the receptors activated by mOSM. Murine OSM is known to signal mainly via the OSMR [28], and to a lesser extent via the LIFR [30,31]. Interestingly, immunoblot analysis revealed higher overall protein levels of OSMR and especially LIFR in OP9 cells compared to NIH/3T3 cells (Figure 2A). It has been demonstrated before, that B cells do not express OSMR and LIFR [32], therefore the pro-B cell line Ba/F3 served as a negative control. Consistent with the immunoblot data, we detected increased *Lifr* mRNA expression in OP9 cells compared to NIH/3T3 cells in microarray analyses, whereas *Osmr* mRNA expression was not altered between both cell lines (Figure 2B). Additionally, flow cytometric analysis revealed strong LIFR surface expression on OP9 cells, whereas LIFR was hardly detectable on NIH/3T3 cells (Figure 2C). These findings suggest the role of the OSM-LIFR signaling axis on the proliferation of OP9 cells.

Proper determination of murine OSMR (mOSMR) surface expression was not possible despite using different flow cytometry antibodies supposed to bind mOSMR (R&D Systems FAB662P, Santa Cruz sc-21797 PE).

We next investigated whether mOSM treatment affects the expression of the OSMR and LIFR in OP9 and NIH/3T3 cells. We detected an initial reduction of total cellular OSMR in both cell lines one hour after mOSM treatment, followed by a return to basal levels after 8 h of mOSM exposure. For the LIFR, a similar regulation pattern expression in OP9 cells was observed with initial downregulation followed by upregulation. In contrast, NIH/3T3 cells retained low-level expression of the LIFR during mOSM treatment (Figure 2D). These results were confirmed by time-course analysis of LIFR surface expression, which was significantly reduced upon mOSM treatment in OP9. Conversely, the mOSM-induced reduction was only slightly detectable in NIH/3T3 cells, possibly due to little LIFR surface expression (Figure 2E). Based on these findings, we conclude that mOSM signals via both OSMR and the LIFR in OP9 cells, whereas NIH/3T3 cells show only marginal LIFR expression and mOSM signaling in this cell line might be largely limited to OSMR. Western Blot quantification is attached in Appendix A.

### 2.3. OSMR Downregulation Attenuates OSM Effects on Proliferation

To investigate the synergistic or antagonistic roles of OSMR or LIFR and mOSM on proliferation, we established an shRNA-mediated knockdown of *Osmr* or *Lifr* in both OP9 and NIH/3T3 cell lines. Knockdown efficacy of four *Osmr*-specific shRNAs and five shRNAs targeting the *Lifr* were evaluated (Figure 3A,B). Exposure times of immunoblots were different between OP9 and NIH/3T3 cells due to lower OSMR and LIFR expression in NIH/3T3 cells. In OP9 cells, targeting *Osmr,* but not *Lifr* partly abolished the mOSM-induced antiproliferative effect compared to cells transfected with an unrelated control shRNA. In contrast, both *Osmr* and *Lifr* knockdown reduced the mOSM-mediated proliferative effect in NIH/3T3 cells (Figure 3C). Our results indicate that the mOSM effects on proliferation are mainly mediated by the OSMR, with only little impact by the LIFR.

In a further step, we analyzed the effect of stable *Lifr* overexpression in both cell lines (Figure 3D). Flow cytometric analysis confirmed enhanced LIFR expression on the surface of *Lifr*-transduced cells (Figure 3E). *Lifr* overexpression attenuated the mOSM-mediated effect on proliferation by approximately 30% and 50% in *Lifr*-overexpressing OP9 and NIH/3T3 cells, respectively (Figure 3F). These results confirm our notion that OSMR is the main mediator of the mOSM effect on proliferation. Taken together, we demonstrate that the proliferative and antiproliferative ability of mOSM depends on signaling through the OSMR. Although mOSM does bind the LIFR, this interaction may only slightly affect cellular proliferation.

### 2.4. OSM Activates JAK-STAT, PI3K-AKT, and MAPK-ERK Pathways in OP9 and NIH/3T3 Cells

After we found mOSM to exert its effects on proliferation through the OSMR in both cell lines, we were then interested in downstream signaling pathway activation and expression. Several studies have indicated that OSM/OSMR mediates its effects via activation of JAK-STAT signaling [4,56]. To study the role of this pathway on proliferation, OP9, and NIH/3T3 cells were treated with mOSM in combination with the JAK1/JAK2-specific inhibitor ruxolitinib at increasing concentrations from 100 nM to 1 µM. In OP9 stromal cells, 1 µM ruxolitinib fully restored the percentage of cells in the S phase in mOSM treated cells to levels of untreated cells, while the proliferative effect of mOSM on NIH/3T3 cells was only partly reversed by ruxolitinib. Treatment with mOSM and 1 µM ruxolitinib reduced the proliferative effect of mOSM in NIH/3T3 cells by 68% (Figure 4A). Ruxolitinib treatment alone did not affect the proliferation of these cell lines (Appendix A). These results suggest that activation of JAK1 and JAK2 is relevant for the mOSM-mediated effect on proliferation in both cell lines.

Immunoblot analysis of the JAK-STAT pathway revealed activation of all JAK family members JAK1, 2, 3, and TYK2 in response to mOSM in both OP9 and NIH/3T3 cells. STAT1, 3, and 5 were also activated in both cell lines. Pretreatment with ruxolitinib prevented the activation of all investigated STAT proteins (Figure 4B). Of note, JAK2 remained phosphorylated in presence of ruxolitinib, most probably in terms of ruxolitinib-induced hyperphosphorylation, which is not associated with JAK2 activity [57]. Total JAK2 and STAT5 levels were higher in OP9 cells, whereas TYK2 was stronger expressed in NIH/3T3 cells. The levels of JAK1, JAK3, STAT1, and STAT3 were not altered between OP9 and NIH/3T3 cells (Figure 4B). Interestingly, transcriptome analysis revealed upregulation of *Stat1* mRNA levels in OP9 and increased expression of *Stat3* and *Stat5* in NIH/3T3 cells in presence of mOSM for 6 h (Figure 4C). These results indicate a similar signaling pathway activation in response to mOSM in OP9 and NIH/3T3 cells. However, differences in the expression and activation of JAK-STAT pathway members by mOSM between OP9 and NIH/3T3 cells might contribute to the differential effects on proliferation.

In addition, to activate the JAK-STAT pathway, OSM was shown to signal via the PI3K-AKT and MAPK-ERK pathways in various cell lines [47,58,59]. Correspondingly, we detected activation of AKT and ERK by mOSM in both OP9 and NIH/3T3 cells. Pretreatment with ruxolitinib prevented the activation of AKT and ERK (Figure 5A). Total ERK levels were elevated in NIH/3T3 compared to OP9 cells. Interestingly, mOSM reduced ERK expression in NIH/3T3, the addition of ruxolitinib reversed this effect. AKT levels were not altered between the cell lines (Figure 5A).

To investigate whether cell cycle regulating proteins are involved in the mOSM-mediated proliferative effect, we examined the tumor suppressor proteins retinoblastoma protein (RB) and p53 in our cell lines. Interestingly, RB was phosphorylated in NIH/3T3 cells and dephosphorylated in OP9 cells upon mOSM treatment, consistent with the mOSM effect on proliferation. However, total RB levels were not altered between OP9 and NIH/3T3 cells before or after OSM treatment (Figure 5B) indicating that the differences in RB activation are a consequence of, but not the cause for the opposing effects of mOSM on proliferation. Western Blot quantification of pRB is attached in Appendix A. In the investigated cell lines, P53 levels were not affected by mOSM (data not shown).

To identify gene and protein classes up- or downregulated upon mOSM treatment we performed functional enrichment analysis of the global transcriptomes of OP9 and NIH/3T3 cells in the presence or absence of mOSM. OP9 cells showed gene set enrichment according to the antiproliferative effect of mOSM: “apoptotic signaling pathway” was positively enriched in response to mOSM, whereas “response to growth factors” and different developmental processes were negatively enriched (Appendix A). In contrast, in NIH/3T3 cells, we detected positive enrichment of “pyrimidine metabolism” and negative enrichment of various differentiation processes upon mOSM treatment (Appendix A). Moreover, functional enrichment analysis revealed that mOSM induces gene sets related to an inflammatory phenotype in both cell lines (“inflammatory response” and “response to interferon-beta”; Appendix A). These results are in line with the well-known role of OSM to support inflammatory processes in various cell lines [4].

### 2.5. IL-6 Has Synergistic Effects to OSM on the Proliferation of NIH/3T3 Cells

We next analyzed mOSM-induced cytokine secretion in OP9 and NIH/3T3 cells, which might affect proliferation in an autocrine manner. For this purpose, supernatants of OP9 cells (+/− mOSM treatment) were added to NIH/3T3 cells and vice versa for 24 h, and cell cycle analysis was performed. OP9 cells treated with the supernatant of mOSM-stimulated NIH/3T3 cells showed impaired proliferative capacity indicated by lower S phase percentage. Conversely, NIH/3T3 cells treated with the supernatant of mOSM stimulated OP9 cells showed increased proliferation (Figure 6A), similar to direct treatment of these cells with mOSM (Figure 1C). We, therefore, assume that the differential effects of the supernatant on the proliferation of the two cell lines are not due to secondary effects of cytokines released in response to mOSM, but rather a direct consequence of residual mOSM in the supernatant.

To analyze the cytokine release of OP9 and NIH/3T3 cells upon mOSM treatment, we performed a bead-based cytokine array of their supernatant. In accordance to other studies [60], we detected a strongly increased IL-6 production in mOSM treated OP9 cells compared to untreated cells (254 pg/mL vs. 7.76 pg/mL, *p* < 0.0001). NIH/3T3 cells also showed increased production of IL-6 upon mOSM treatment (13.9 pg/mL vs. not detectable), however significantly less than OP9 cells (Figure 6B). This was corroborated by transcriptome analysis revealing upregulation of IL-6 mRNA levels in OP9 and NIH/3T3 + mOSM (Appendix A). In contrast, the release of the monocyte-attracting chemokine MCP-1/CCL2 was induced by mOSM in NIH/3T3, but not OP9 cells (Figure 6B). IL-10 and IL-27 levels were not significantly altered after mOSM treatment in both cell lines (Appendix A).

We next studied the proliferative effect of mIL-6 on both cell lines and found that mIL-6 promoted the proliferation of NIH/3T3 cells (1.8-fold), albeit to a lesser extent than mOSM (2.7-fold). The proliferation of OP9 cells was not affected by mIL-6 (Figure 6C). Interestingly, phosphorylation-specific flow cytometry revealed STAT3 activation in NIH/3T3 but not in OP9 cells upon mIL-6 treatment (Figure 6D). To investigate the IL-6-mediated effects on NIH/3T3 cells further, we treated them with mOSM or mIL-6 in the presence or absence of neutralizing IL-6 antibodies. While treatment of NIH/3T3 cells with IL-6 neutralizing antibodies reversed the mIL-6-induced proliferation, it did not affect the OSM-induced proliferation (Figure 6E). These results suggest that the IL-6 secretion in response to mOSM is not sufficient by itself to affect the proliferation of NIH/3T3 cells and that mIL-6 and mOSM have synergistic effects on the proliferation of this cell line.

## 3. Discussion

In the present study, we investigated the OSMR dependent regulation of proliferation in OP9 and NIH/3T3 cells by mOSM. In line with previous reports [55,61], we uncovered mOSM as a negative regulator of proliferation of BM stromal cells and as a stimulator of proliferation in NIH/3T3 fibroblasts. OSMR and LIFR were differentially expressed between OP9 and NIH/3T3 cells. In addition, the expression of both receptors was altered upon mOSM treatment. Further investigations identified the OSMR as the key receptor in mOSM-mediated effects on proliferation.

The contribution of OSM in tumorigenesis has been controversial: Depending on the tumor entity, OSM was shown to promote tumor progression and epithelial-mesenchymal transition [44,45,46,47,62] or prevent tumor cell growth [1,40,41,42,43,49]. Our study investigated these opposing effects on proliferation as a part aspect of tumorigenesis in two defined in vitro models. When we analyzed the receptor composition of both cell lines, we detected higher overall protein levels of OSMR and especially LIFR in OP9 cells compared to NIH/3T3 cells. In addition, mOSM induced an initial downregulation of OSMR and LIFR in OP9 cells and of the OSMR in NIH/3T3 cells. We assume receptor downregulation is a consequence of receptor internalization after ligand binding followed by a compensatory enhancement of receptor expression. In subsequent knockdown experiments, we found that mOSM affects cell proliferation of both cell lines primarily by signaling through the OSMR, thus confirming previous data identifying mOSM as an activator and high-affinity binding partner of OSMR [28,29]. Studies of the huOSM system support the idea of an OSMR dependent regulation of proliferation by OSM [48].

Our study described an inferior role for the LIFR in the regulation of cellular proliferation by mOSM. *Lifr* overexpression was associated with attenuated effects on proliferation by mOSM in both cell lines, indicating that *Lifr* overexpression might act to decrease levels of free mOSM for binding to the OSMR leading to reduced effects on proliferation. In support of this hypothesis, general receptor-ligand studies showed that increased receptor surface density was associated with increased ligand occupancy [63]. Furthermore, *Lifr* knockdown attenuated the OSM-mediated response on proliferation in NIH/3T3 but not in OP9 cells. We assume that *Lifr* knockdown might have been more efficient in NIH/3T3 cells due to *per se* lower LIFR expression. Consequently, Lifr knockdown impaired the mOSM effect on proliferation in NIH/3T3 cells. In contrast, OP9 cells showed higher LIFR expression compared to NIH/3T3, therefore *Lifr* knockdown possibly did not sufficiently impair LIFR expression to detect altered mOSM effects on proliferation. Whereas our study revealed only marginal effects of mOSM-LIFR interaction on proliferation, recent studies identified the LIFR to be important in mOSM-regulated bone metabolism [30,31]. Therefore, mOSM-LIFR signaling might be dependent on the specific cellular context, similar to mOSM-OSMR signaling [28] and might exert more effects on metabolism than proliferation.

As previous research showed that huOSM affects the proliferation of chondrosarcoma and lung adenocarcinoma cells in a JAK-dependent manner [41,56], we aimed to define the exact roles of these signaling molecules in the mOSM-dependent regulation of proliferation. We used the JAK1 and JAK2 inhibitor ruxolitinib, which is approved for the treatment of myelofibrosis [64], polycythemia vera [65] and acute graft-versus-host disease [66], where it primarily exerts cytoreductive and antiproliferative effects. Interestingly, we found that ruxolitinib completely abolished the mOSM-mediated antiproliferative effect in OP9 cells. In line with this finding, another study identified ruxolitinib as an agent promoting hair cell growth [67]. These results indicate that ruxolitinib may exert proliferative capacities in certain cellular contexts, which should be taken into account in clinical studies with systemic use of this inhibitor.

In NIH/3T3 cells, ruxolitinib reduced OSM-induced proliferation only partly, indicating the involvement of additional mechanisms. These might include an altered abundance of signaling proteins, as we detected increased levels of total TYK2 and ERK1/2 in NIH/3T3 cells, whereas OP9 cells contained higher levels of total JAK2 und STAT5 proteins.

Downstream of JAK activation, we found phosphorylation of STAT1, 3, and 5 in both cell lines in response to mOSM treatment. Interestingly, transcriptome analysis revealed upregulation of *Stat1* mRNA levels in OP9 and increased expression of *Stat3* and *Stat5* in NIH/3T3 cells. Active STAT3 and STAT5 signaling are known to increase proliferation in different cellular contexts [68,69,70,71], while STAT1 mainly acts as an antiproliferative factor [72,73,74]. Induction of proliferation in response to huOSM was also demonstrated to be related to STAT3 activation [47], whereas growth-inhibitory effects of huOSM have been found to be mediated by JAK3 and STAT1 activation [41]. Another study demonstrated that both induction and inhibition of proliferation elicited by huOSM were STAT3-dependent [48]. Based on these observations, we propose that responses to mOSM might depend on specific balances and ratios of STAT proteins and their interaction with co-activators. Indeed, previous research revealed evidence for cross-regulation between different STAT proteins [75,76].

In our study, ERK expression was found to be lower in OP9 compared to NIH/3T3 cells. Interestingly, mOSM induced downregulation of ERK expression in NIH/3T3 whereas OP9 cells retained constant ERK levels. This supports previous research showing OSM-mediated downregulation of ERK expression that correlated with increased proliferation [77].

Transcriptome analysis of OP9 and NIH/3T3 cells highlighted enrichment of specific gene sets corresponding to the antiproliferative and proliferative effects of mOSM. This result was corroborated by differential RB activation in OP9 versus NIH/3T3 cells. Furthermore, another study showed downregulation of the cyclin D1/CDK4 complex and reduced pRB levels upon OSM stimulation inducing growth arrest in mouse skeletal muscle cells [78].

Future studies should further investigate the differences in signaling proteins using specific inhibitors and knockdown experiments. Moreover, OSM was shown to activate c-Jun N-terminal kinase (JNK), p38 mitogen-activated protein kinase [36] and protein kinase C-delta (PKCd) pathways [37,38,39]. Two studies demonstrated, that OSM stimulated the proliferation of prostate cancer cells [45] and ovarian cancer cells [47] by activation of the p38 mitogen-activated protein kinase pathway. We propose to analyze and compare the activation and expression of the mentioned pathways which might help to identify additional mechanisms leading to the opposing effects of mOSM on proliferation.

We detected mOSM-induced IL-6 secretion in OP9 and NIH/3T3 cells, which is consistent with previous reports showing IL-6 release in response to OSM in different contexts [4,60]. Interestingly, m-IL6 promoted the proliferation of NIH/3T3 cells but did not affect the proliferation of OP9 cells. STAT3 was not phosphorylated in OP9 cells upon mOSM treatment suggesting that these cells lack IL-6 receptor expression. However, neutralizing IL-6 antibodies could not reverse the mOSM proliferative effect in NIH/3T3 cells. The measured IL-6 concentration in the supernatant of NIH/3T3 + mOSM (0.0139 ng/mL) was approximately 0.01% of the IL-6 concentration used for cell cycle analysis (10 ng/mL), suggesting that this IL-6 concentration was not sufficient to affect proliferation. Moreover, we assume that the pro-inflammatory cytokine IL-6 [4] contributes to the mOSM-generated inflammatory phenotype indicated by gene set enrichment analysis in both cell lines. IL-6 is a member of the “inflammatory response” gene set [79] and transcriptome analysis of OP9 and NIH/3T3 cells + mOSM revealed elevated IL-6 mRNA levels.

Of note, findings from studies on mOSM might not be transferred completely to the structural different huOSM. In contrast to mOSM, huOSM was shown to bind to the type I and type II OSM-receptors with high affinity [26,27]. While huOSM is able to activate huOSMR, huLIFR, and mLIFR, mOSM does not activate any of the human receptors [28]. This suggests that to some extent different pathways might be activated by mOSM and huOSM.

Taken together, our study has demonstrated that both proliferative and antiproliferative effects of mOSM were mediated mainly through the OSMR. Based on our findings, mOSM seems to bind the LIFR with marginal but synergistic (compared to the OSMR) effects on proliferation. Furthermore, mOSM induces activation of the JAK-STAT, PI3K-AKT, and MAPK-ERK pathways in OP9 and NIH/3T3 cells with differences in total protein levels between the two cells lines. RB activation and gene set enrichment clusters correspond with the proliferative or antiproliferative response to mOSM.

Due to its multifaceted activities in various cancer entities, OSM is currently considered a novel therapeutic target, and a number of targeting antibodies and other compounds have been developed in preclinical research [80]. Our findings indicate that a more comprehensive understanding of how OSM regulates processes promoting tumor progression (such as aberrant proliferation, inflammation, modulation of the immune system, and metastasis) is required as a prerequisite to the initiation of clinical trials targeting this cytokine.

## 4. Materials and Methods

### 4.1. Cell Culture

Ba/F3 cells and NIH/3T3 cells were obtained from the German Resource Centre for Biological Material (DSMZ). Phoenix E helper-virus free ecotropic packaging cells were a kind gift from G. Nolan, Stanford, USA. OP9 and M2-10B4 cells were a kind gift from Christine Dierks, University Hospital Freiburg, Germany. EL08-1D2 cells were kindly provided by R. A. Oostendorp, Munich, Germany. Primary murine stromal cells were isolated from C57BL/6 mice. Primary human stromal cells from healthy donors have been obtained from the stem cell transplant department of the University of Freiburg, Germany.

Ba/F3 cells were maintained in RPMI 1640 medium (Thermo Fisher, Waltham, MA, USA) containing 10% fetal calf serum (FCS; Sigma-Aldrich, St. Louis, MO, USA), 1% Penicillin-Streptomycin (PS; Thermo Fisher), and 2 ng/mL murine interleukin-3 (IL-3; Peprotech, Cranbury, NJ, USA). Phoenix E and NIH/3T3 cells were maintained in DMEM (Thermo Fisher) supplemented with 10% FCS. NIH/3T3 cells were cultivated in presence of 1% PS. OP9 cells and M2-10B4 cells were maintained in alpha-MEM (Thermo Fisher) supplemented with 10% FCS and 1% PS. EL08-1D2 cells were maintained in alpha-MEM supplemented with 7.5% FCS, 2.5% horse serum (Thermo Fisher), 1% PS und 200 µL ß-Mercaptoethanol (50 mM; Thermo Fisher). All cells were cultured at 37 °C with 95% O_2_, 5% CO_2_.

### 4.2. Cell Culture Supplements

Murine recombinant OSM was purchased from Cell Signaling (Danvers, MA, USA). Human recombinant OSM and murine recombinant IL-6 were purchased from Peprotech. Cytokines were used at a final concentration of 10 ng/mL. Ruxolitinib was purchased from ChemieTek, Indianapolis, IN, USA, the neutralizing anti-IL-6 antibody (MM600C) from Thermo Fisher.

### 4.3. Retrovirus Production and Transduction

Phoenix E cells were transiently transfected using Turbofect (Thermo Fisher) and retroviral stocks were collected twice at 12-h intervals beginning 24 h after transfection. OP9 and NIH/3T3 cells were stably transduced with retroviral supernatant supplemented with 4 µg/mL polybrene (Sigma Aldrich, Saint Louis, MO, USA) for 3 times every 12 h.

### 4.4. Proliferation Assays

Proliferation was assessed using Click-iT^®^ EdU Alexa Fluor^®^ 647 Flow Cytometry Assay Kit (Thermo Fisher) according to the manufacturer’s instruction. DNA content was quantified using FxCycle™ Violet Stain (Thermo Fisher) according to the manufacturer’s instruction. EdU was used at a concentration of 20 µM.

Cells were treated for 24 h with mOSM or huOSM, respectively. EdU was added during the last hour of treatment or the last 12 h of treatment for primary stromal cells. For experiments comparing OP9 and NIH/3T3 cells, cells were serum-starved with 1% FCS for six hours before and during treatment. Ruxolitinib was added at the same time as mOSM for proliferation assays.

### 4.5. DNA Constructs and Knockdown

Murine wildtype *Lifr* cDNA (MR221059) was obtained from OriGene (Rockville, MD, USA). The retroviral vector was constructed by cloning the *Lifr* cDNA into the MigRI retroviral vector coexpressing the enhanced green fluorescent protein (EGFP; a kind gift from W. Pear, Philadelphia, PA, USA) [81]. Empty MigRI was used as a control for *Lifr* overexpression. RIEP (rtTA-IRES-EcoReceptor-PGK-PuroR), TREBAV (TRE-dsRed-miRE-PGK-BSDr-2A-Venus), and pSuper plasmids were kindly provided by C. Miething, Freiburg, Germany. For doxycycline-inducible knockdown of *Lifr* or *Osmr* respectively, OP9 and NIH/3T3 cells were stably transduced first with RIEP and subsequently with TREBAV vectors containing the appropriate shRNA. All cell lines including control cell lines were cultured in presence of 1 µg/mL doxycycline. Doxycycline was added at least 48 h before starting experiments. The following shRNA sequences were used for the knockdown experiments: *Lifr* 5‘ TGCTGTTGACAGTGAGCGCTACCATGTTGCTGTAGACAAATAGTGAAGCCACAGATGTATTTGTCTACAGCAACATGGTAATGCCTACTGCCTCGGA3‘ (here named: *Lifr* 1), *Osmr* 5′ TGCTGTTGACAGTGAGCGCAAGCATCTTCTTGTAAACTTATAGTGAAGCCACAGATGTATAAGTTTACAAGAAGATGCTTTTGCCTACTGCCTCGGA3‘ (here named: *Osmr* 5) and *Renilla (control)* 5′ CGCGACTCCTATAATTTCTAATTAGTGAAGCCACAGATGTAATTAGAAATTATAGGAGTCGCT3′. Further shRNA sequences are described in Appendix A. All constructs were verified by sequencing.

### 4.6. Flow Cytometric Analysis and Cell Sorting

Flow cytometry was performed using an LSRFortessa^TM^ (Becton Dickinson, Franklin Lakes, NJ, USA) cytometer and FlowJo 10.4 was used to analyze the data.

Antibodies used were: LIFRalpha (FAB5990P; R&D Systems, Minneapolis, MN, USA) and pSTAT3 (557815, Becton Dickinson). Cells were sorted on an Aria III^TM^ (Becton-Dickinson) as described before [82,83]. Antibodies used for cell sorting were CD45 (25-0451-82; Thermo Fisher), Ter-119 (116222; Biolegend, San Diego, CA, USA), Gr-1 (108416, Biolegend), CD3 (25-0031-82, Thermo Fisher) and CD11b (25-0112-82, Thermo Fisher). CD31 (102422, Biolegend) Sca-1 (108126, Biolegend), CD166 (12166182, eBioscience, San Diego, CA, USA) and CD140 (135906, Biolegend).

### 4.7. Immunoblotting

Immunoblot analysis was performed as previously described [84]. Cells were lysed in lysis buffer containing 10 mM Tris (tris(hydroxymethyl) aminomethane)–HCl (pH 7.5), 130 mM NaCl, 5 mM EDTA (ethylenediaminetetraacetic acid), 1% Triton^TM^ X-100, 20 mM sodium phosphate (pH 7.5), 10 mM sodium pyrophosphate (pH 7.0), 50 mM NaF, 1 mM sodium orthovanadate, 1 mM glycerolphosphate, and protease inhibitors (Roche Diagnostics, Basel, Switzerland). Blotting was performed on polyvinylidene fluoride (PVDF) membranes (Immobilon-P; Merck Millipore, Billerica, MA, USA). Antibodies against pAKT (cs9271), AKT (cs9272), pERK1/2 (cs9101), ERK1/2 (cs4695), pJAK2 (cs4406), JAK2 (cs3230), pJAK3 (cs5031), JAK3 (cs8863), pRB (cs8516), pSTAT1 (cs7649), pSTAT3 (cs9131), STAT3 (cs9132), pSTAT5 (cs9314), STAT5 (cs94205) and pTYK2 (cs9321) were purchased from Cell Signaling. TYK2 (sc5271), LIFR (sc515337), OSMR (sc376511), RB (sc50) and Vinculin (sc73614) were from Santa Cruz, Dallas, TX, USA. pJAK1 (44-422G) was from Thermo Fisher. STAT1 (21120) was from Becton Dickinson. ß-actin (A5316) was from Sigma. Vinculin and ß-actin were used as loading controls. Secondary antibodies used were mouse IgG HRP-linked (cs7076) and rabbit IgG HRP-linked (cs7074) from Cell Signaling. Blots were developed using SuperSignal^TM^ chemoluminescent substrates (Thermo Fisher). Quantification of Western Blots was done using LabImage 1D Sofware (Intas Science Imaging, Göttingen, Germany).

### 4.8. Cytokine Array

Cytokine secretion was measured using LEGENDplex™ Mouse Inflammation Panel (Biolegend) according to the manufacturer’s instruction. The cytokine array was performed using an LSRFortessa^TM^ (Becton Dickinson, Franklin Lakes, NJ, USA) cytometer.

### 4.9. RNA Isolation and Microarray

RNA was isolated with the RNeasy Mini Kit (Qiagen) and transcribed into cDNA with the First Strand cDNA Synthesis Kit (Thermo Fisher). RNA quality was assayed using an Agilent 4200 TapeStation (Agilent Technologies, Santa Clara, CA, USA). Mouse Clariom S Assay (Thermo Fisher) was used to analyze mRNA expression according to the manufacturer’s instructions. The *p*-values were calculated by Transcriptome Analysis Console (Affymetrix) software. Data were analyzed using Metascape analysis (Metascape A Gene Annotation and Analysis Resource. Available online: http://metascape.org/, accessed on 21 October 2021) [85] with gene symbols from genes with >2-fold upregulation or downregulation and *p* < 0.05 in control cells compared to mOSM treated cells. Microarray data have been deposited to the Gene Expression Omnibus (GEO) and will be accessible through GEO Series accession number GSE185646.

### 4.10. Statistical Analysis

Statistical analysis was performed using GraphPad Prism software. *p*-Values were determined by Student’s *t*-test and one-way ANOVA with Bonferroni post-test for comparison of more than two groups. Data are represented as means ± SEM.

## Figures and Tables

**Figure 1 ijms-22-11649-f001:**
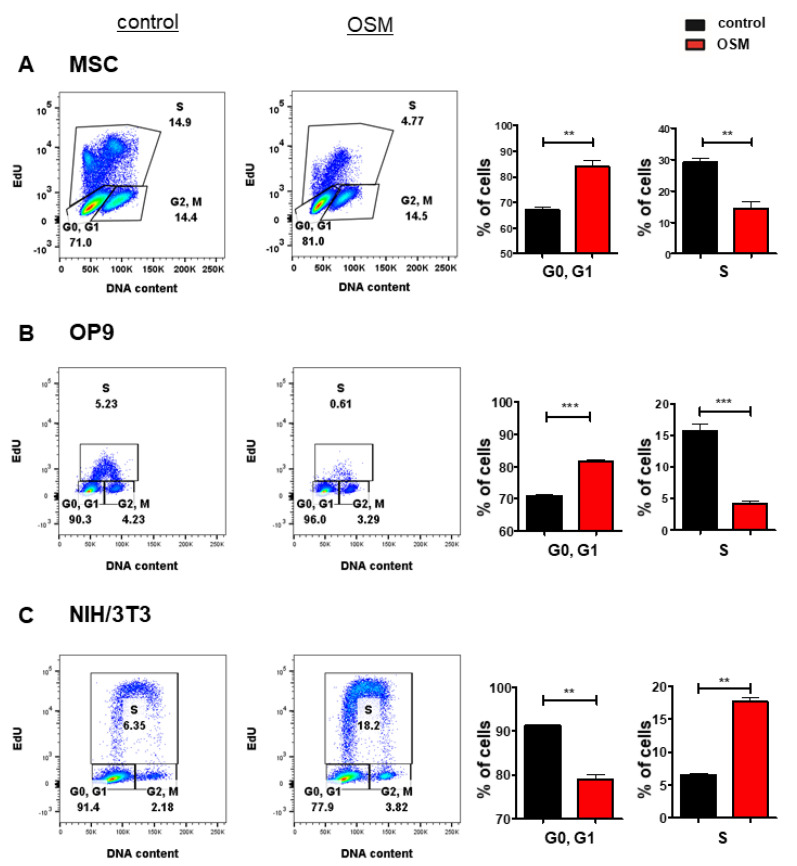
OSM has differential effects on the proliferation of BM stromal cells and fibroblasts. Representative FACS plot (left) and quantification of cell cycle phases (right) of (**A**) primary murine BM stromal cells and (**B**) OP9 cells and (**C**) NIH/3T3 cells in presence or absence of 10 ng/mL mOSM. For A, cells were cultured in a medium containing 10% FCS and treated with mOSM for 24 h. For B and C, cells were serum-starved for 6 h and treated with mOSM for 24 h. The proliferation was assessed using EdU incorporation. Cells were exposed with EdU during the last hour of treatment (**B**,**C**) or the last 12 h of treatment (**A**). The DNA content was quantified using FxCycle^TM^ Violet Stain. Student’s unpaired *t*-test. ** *p* < 0.01, and *** *p* < 0.001.

**Figure 2 ijms-22-11649-f002:**
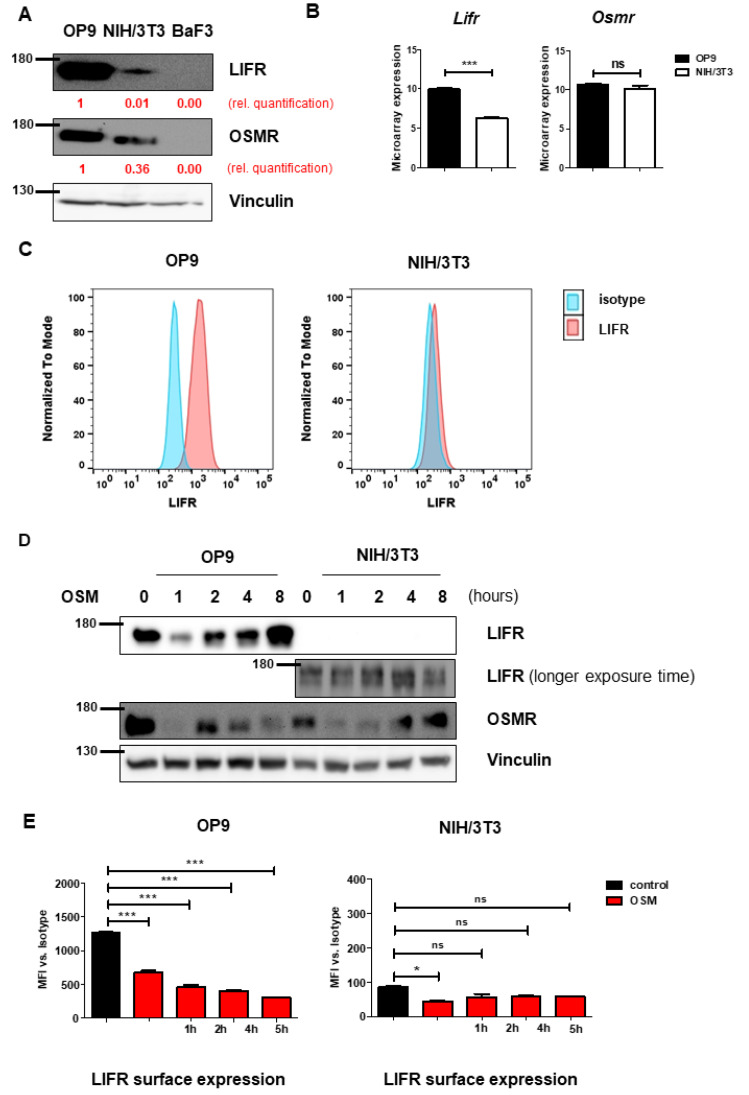
OP9 and NIH/3T3 cells show differential levels of OSMR and LIFR. (**A**) OP9 and NIH/3T3 cells were examined for LIFR and OSMR expression. BaF3 cells were used as a negative control. Protein molecular weight is labeled in kDa. Relative protein quantities are marked in red. (**B**) Microarray analysis of OP9 and NIH/3T3 cells showing *Osmr* and *Lifr* expression. The *p*-values were calculated by Transcriptome Analysis Console software. *** *p* < 0.001. ns = not significant. (**C**) OP9 and NIH/3T3 cells were analyzed for LIFR surface expression using FACS. (**D**) OP9 and NIH/3T3 cells were examined for expression of LIFR and OSMR in relation to different incubation periods with 10 ng/mL mOSM. Protein molecular weight is labeled in kDa. (**E**) OP9 and NIH/3T3 cells were examined for LIFR surface expression in relation to different incubation periods with 10 ng/mL mOSM. *p*-Values were calculated using one-way ANOVA and Bonferroni post-test. * *p* <0.05 and *** *p* < 0.001. ns = not significant.

**Figure 3 ijms-22-11649-f003:**
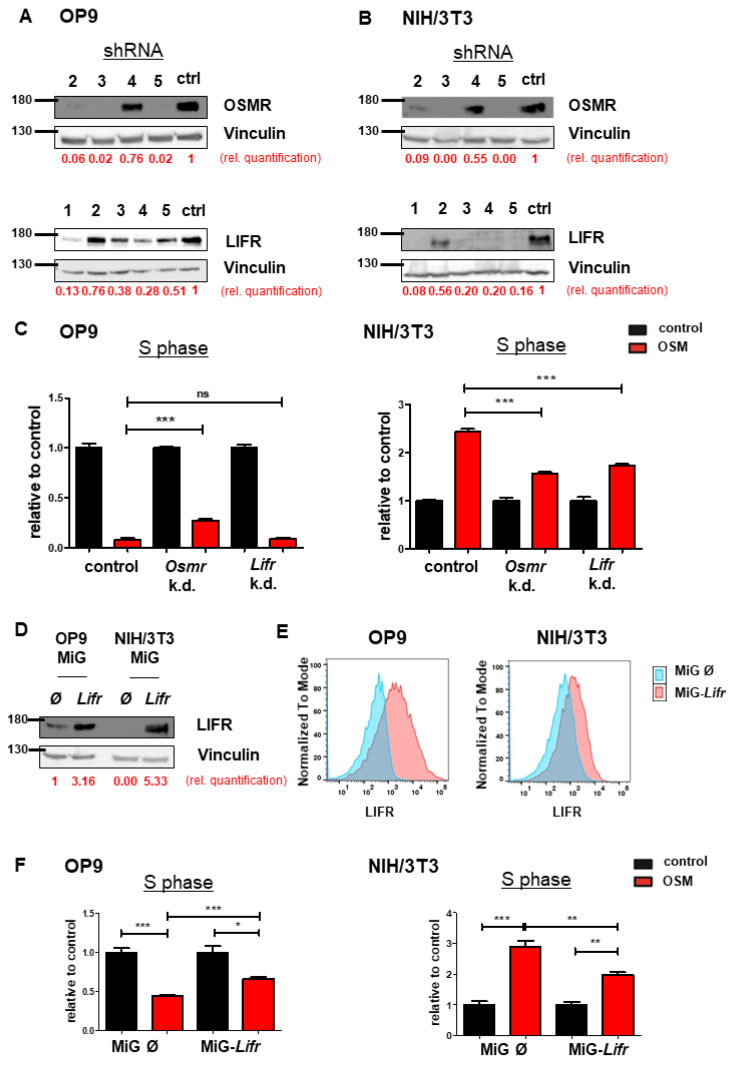
OSMR downregulation attenuates OSM effects on proliferation. (**A**,**B**) Immunoblot quantifying the shRNA-induced knockdown of *Osmr* (n = 4) and *Lifr* (n = 5) in (**A**) OP9 and (**B**) NIH/3T3 cells. The shRNAs *Osmr* 5 and *Lifr* 1 were used in subsequent experiments. Protein molecular weight is labeled in kDa. Relative protein quantities are indicated in red. (**C**) Relative quantification of S phase of OP9 and NIH/3T3 cells carrying a knockdown for either *Osmr*, *Lifr,* or a *Renilla* control in presence or absence of 10 ng/mL mOSM. Cell cycle analysis was performed as described before. One-way ANOVA and Bonferroni post-test. *** *p* < 0.001. ns = not significant. (**D**) Immunoblot quantifying the LIFR overexpression in OP9 and NIH/3T3 cells. Protein molecular weight is labeled in kDa. Relative protein quantities are indicated in red. (**E**) *Lifr* overexpressing OP9 and NIH/3T3 cells were examined for LIFR surface expression compared to cells carrying the mock control. (**F**) Relative quantification of S phase of OP9 and NIH/3T3 cells transduced with *Lifr* or empty vector (MiG). Cell cycle analysis was performed as described before. One-way ANOVA and Bonferroni post-test. * *p* < 0.05, ** *p* < 0.01, and *** *p* < 0.001.

**Figure 4 ijms-22-11649-f004:**
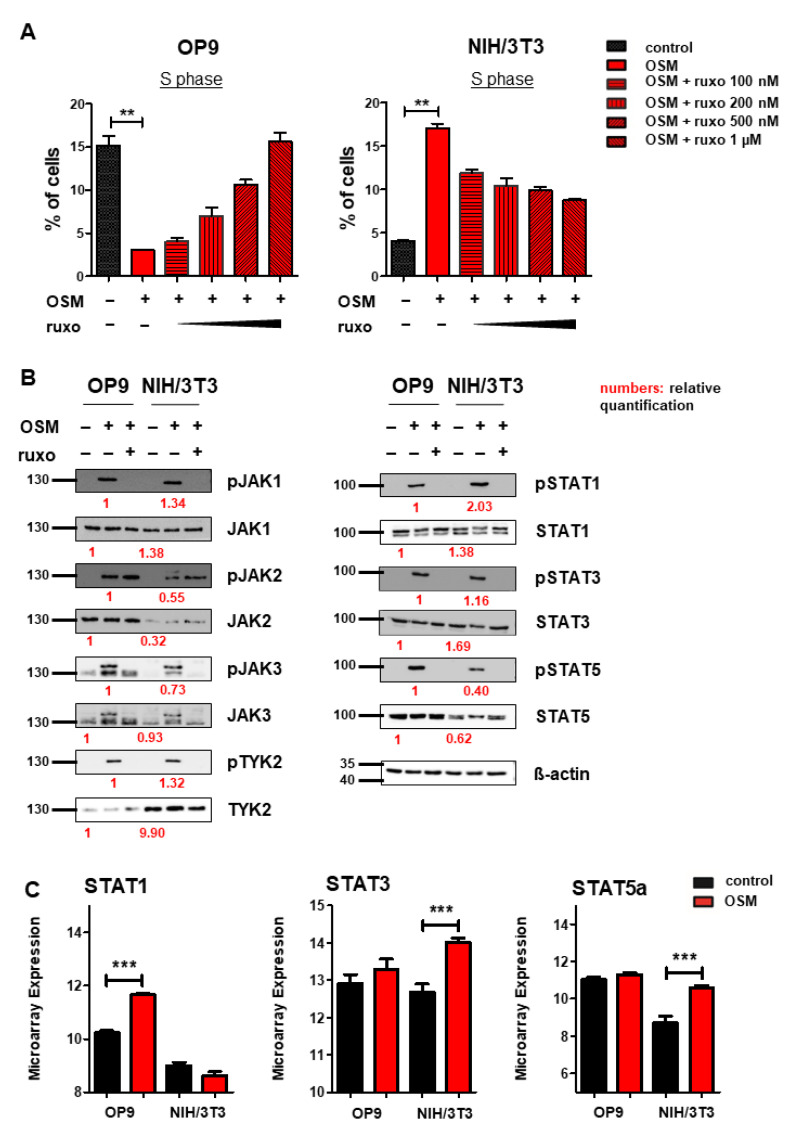
OSM activates the JAK-STAT pathway in OP9 and NIH/3T3 cells. (**A**) Quantification of S phase using an EdU incorporation assay. Cells were serum-starved for 6 h and treated for 24 h with 10 ng/mL mOSM +/− ruxolitinib (ruxo) at indicated concentrations. Student’s unpaired *t*-test. ** *p* < 0.01. (**B**) OP9 and NIH/3T3 cells were used to examine the expression and activation of the JAK-STAT pathway at 10 ng/mL mOSM +/− 1 µM ruxolitinib. Cells were serum-starved overnight. Ruxolitinib was added for 2 h and mOSM 15 min before harvesting the cells. Protein molecular weight is labeled in kDa. Relative protein quantities are indicated in red. (**C**) Microarray analysis of OP9 and NIH/3T3 cells showing *STAT1, 3, and 5a* expression. *** *p* < 0.001.

**Figure 5 ijms-22-11649-f005:**
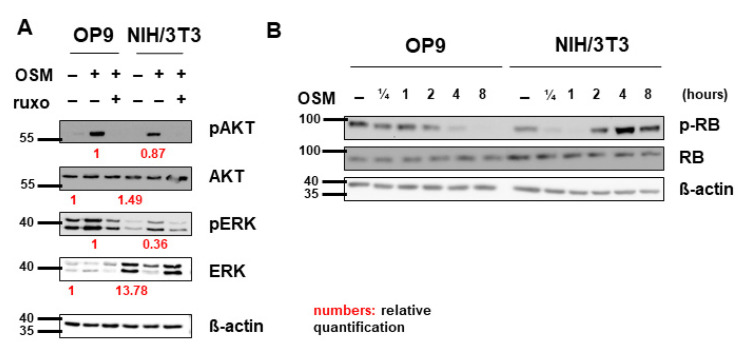
Pathway activation patterns were consistent with the OSM effect on proliferation. (**A**) OP9 and NIH/3T3 cells were used to examine the expression and activation of the MAPK-AKT and PI3K-ERK pathways at 10 ng/mL mOSM +/− 1 µM ruxolitinib. Cells were serum-starved overnight. Ruxolitinib was added after 2 h and mOSM 15 min before harvesting the cells. Protein molecular weight is labeled in kDa. Relative protein quantities are indicated in red. (**B**) OP9 and NIH/3T3 cells were used to examine the expression and phosphorylation of retinoblastoma protein (RB) by mOSM for different time periods. Protein molecular weight is labeled in kDa.

**Figure 6 ijms-22-11649-f006:**
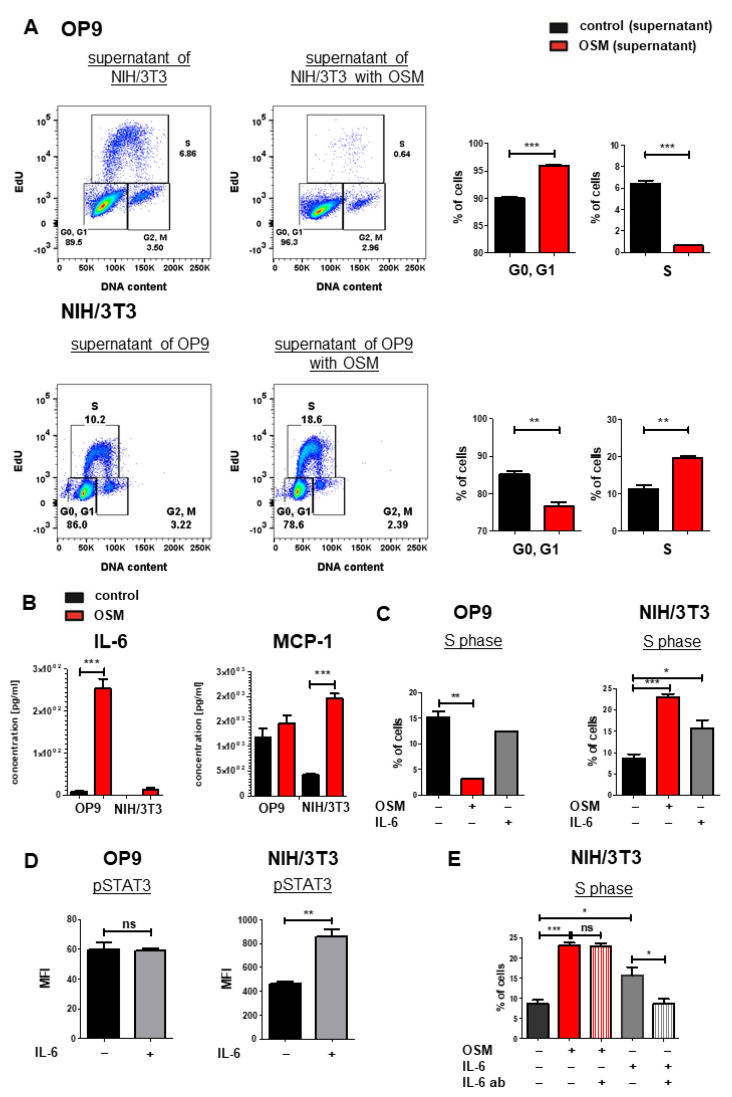
IL-6 has synergistic effects to OSM on the proliferation of NIH/3T3 cells. (**A**) Representative FACS plot (left) and quantification of cell cycle phases (right) of OP9 and NIH/3T3 cells with supernatant from the other respective cell line. Cells were treated with 10 ng/mL mOSM. Student’s unpaired *t*-test. ** *p* < 0.01, and *** *p* < 0.001. (**B**) Cytokine secretion of OP9 and NIH/3T3 cells (+/- mOSM treatment) was measured using a bead-based array detecting 13 cytokines. Student’s unpaired *t*-test *** *p* < 0.001. (**C**) Quantification of the S phase for OP9 and NIH/3T3 untreated or treated with 10 ng/mL mOSM or mIL-6. Student’s unpaired *t*-test. * *p* < 0.05, ** *p* < 0.01, and *** *p* < 0.001. (**D**) Phospho-specific STAT3 flow cytometry analysis of OP9 and NIH/3T3 cells +/− 10 ng/mL IL-6. *** p* < 0.01. ns = not significant. (**E**) Quantification of S phase for NIH/3T3 cells treated with 10 ng/mL mOSM +/− IL-6-ab or 10 ng/mL mIL-6 +/− IL-6 ab. One-way ANOVA and Bonferroni post-test. * *p* < 0.05 and *** *p* < 0.001. ns = not significant.

## Data Availability

Publicly available datasets were analyzed in this study. This data can be found here Gene Expression Omnibus. Available online: https://www.ncbi.nlm.nih.gov/geo/query/acc.cgi?acc=GSE185646, Accession number: GSE185646 (accessed on 21 October 2021).

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
