# Peer review of "Murine Oncostatin M Has Opposing Effects on the Proliferation of OP9 Bone Marrow Stromal Cells and NIH/3T3 Fibroblasts Signaling through the OSMR"

_ijms, 2021, doi:10.3390/ijms222111649_

Round 1

Reviewer 1 Report

The abstract and the introduction need to be more focused. It is pretty hard to follow why the authors decide to select these cell lines and after the first reading, this choice seemed to almost randomly make. Example: In line 20 they claim that the role of OSM in tumorigenesis is still controversial, however, they don't perform experiments with tumors cells. They could also better discuss in which tumor cells the OSM has proliferative vs. anti-proliferative effects and make a connection with their cells choices/results (like the tentative made in line 335).

-The data presented in the paper are not compelling: Figure 1 explains better why OP9 and NIH3T3 are starved and then treated.

-The proliferation effect should be evaluated also with cells count at 48 and 72h to observe an actual increase in the cell number. Moreover, what is the apoptotic rate of the cells after OSM treatment?

-Is the duplication time of MSC/OP9/NIH3T3 cells similar?

Figure 2: In the text should be mentioned why you use BaF3 as a negative control.

Figure 3: The amount of OSMR and LIFR in OP9 and NIH3T3 observed in the WB of panel 2A doesn't seem to correspond to what was observed in the control of Figures 3A and B. A WB quantification could help. 

Figure 4: I guess it is important to include the Ruxo treatment alone since this drug alters by itself the proliferation rate of several cell lines. 

Reviewer 2 Report

The manuscript of Jacob et al. describes the effect of Oncostatin M on the regulation of cell cycle progression and proliferation in different cell types. They found that OSM promotes proliferation of OP9 and NIH/3T3 fibroblasts and increases of the number of cells in S phase compared to untreated cells. Only OP9 cells showed a reduction of cells that remained in the G0/G1 phase after OSM treatment. Next, they investigated the mechanism that is responsible for these changes. They found that both OSMR and LIFR receptors have an important role in proliferation by using gain and loss of function models. Finally, they investigated the involvement of different parts of the JAK-STAT signaling pathway with a chemical inhibitor.Overall, the study is well conducted, although the actual difference in cellular signaling that explains the differential effects on proliferation between the cell-lines remains to be elucidated (inhibition of OSM signaling reduces the positive or negative effect of OSM on proliferation).

Main comments:

Is there a dose-response effect of OSM concentration on the pro and anti-proliferative effect in these cell-lines?

Why did the authors culture the three cell-lines they intend to compare in RPMI, alpha-MEM and DMEM? The reviewer appreciates the medium swap, but would the effect be the same if cells were cultured in other media for prolonged time (weeks/months)?

How is it possible that there is no differential effect of OSMR/LIFR gain and loss of function in the cell-lines? It would appear that OSM signaling is reduced in both OP9 and NIH/3T3 cells, which results in more proliferation when OSM inhibits this and less proliferation when OSM promotes is. As such, there must be another OSM-specific downstream signaling event that differs between these cell-lines. Can the authors pinpoint another regulatory mechanism aside from JAK/STATAKT/ERK or pRB based on literature? And make a recommendation for future studies, to suggest how the underlying reason for this differential effect can be elucidated?

Immunoblot data; 1) Please add appropriate molecular weight markers to immunoblot images throughout the manuscript. 2) Quantification of immunoblots would aid the interpretation of the results.

In general, there is quite some discussion in the results section that should be reserved for the discussion section. E.g. line 117-119, 152/153 and 187/188.

Line 404/405; “how OSM positively or negatively regulates proliferation in different cell types is required as a prerequisite to the initiation of clinical trials targeting this cytokine.

Do the authors consider cancer as a mere proliferation disease? OSM could very well function in other hallmarks of cancer (Hannahan & Weinberg), such as tumor promoting inflammation and/or activating invasion and metastasis.

Line 497; It is customary that the data from whole transcriptome analysis is made publicly available, but there is no geo accession number. Will the authors make this data publicly available?

Minor comments;

Why did the authors choose to continue with only one shRNA? It would have been elegant if the consequence of knock-down was independently validated with another shRNA.

Line 184/186; “Ruxolitinib treatment alone did not affect the proliferation of these cell lines (data not shown).” In the opinion of the reviewer it would be relevant to show these data in the supplement.

Line 196: “The levels of JAK1, JAK3, STAT1 and STAT3 were not altered between OP9 and NIH/3T3 cells (Figure 4B).” In the opinion of the reviewer, JAK3 also appears to be reduced. Quantification would aid in assessing the total expression levels.

Is there a difference in proliferation speed between the investigated cell-lines? NIH/3T3 cells are known to proliferate fast.

The use of huOSM and mOSM should be avoided by utilizing OSM (all capital) for human and Osm (first letter in capital) for murine genes and proteins throughout the manuscript.

Methods

Line 439; the reviewer assumes that the authors mean 8 µg/ml and nog g/ml polybrene?

Line 467; please add the sequence of the control shRNA in the main manuscript and state here what kind of control virus (empty MigRI?) was used for LIFR overexpression.

Line 483; “Immunoblot analysis was performed as previously described [79].” This reference refers to a paper from the year 2003, were there no changes to the protocol? A basic summary should at least be provided that covers the lysis buffer, membrane type, secondary antibodies and method of signal detection.

Line 494/495; What device was used to measure this?

Textual;

Line 17: ‘’The IL-6 cytokine Oncostatin’’ à Oncostatin M is a member of the IL-6 cytokine family

Line 99: 2.4 à 2.2?

Line 138: 2.5 à 2.3?

Line 140: an à a?

Line 142: was à were?

Line 173: 2.2 à 2.4?

Line 250: 2.3 à 2.5?

Line 449: sometimes pre-incubation was used à mentioned in line 191

Round 2

Reviewer 1 Report

The authors have addressed my major revisions.